# Universal approximation capabilities of coherent diffractive systems

**Lennart Schlieder**         Valentin Volchkov         Alexander Song         Peer Fischer
Bernhard Schölkopf

## Abstract

Coherent optical computing systems are a promising avenue to increasing computation speed and solving energy requirements for machine learning applications. These systems leverage the diffraction of coherent waves to perform calculations in the optical domain. Although diffraction is inherently a linear process in complex space $\mathbb{C}$, empirical results show that these systems can outperform standard linear matrix multiplications in $\mathbb{R}$, because photo-sensors project from complex space to real space. Here we provide theoretical insights to explain this phenomenon. We demonstrate that a system consisting of multiple phase-plates, two output photo-detectors, and the appropriate input encoding is theoretically able to learn any one-dimensional function. Additionally, we show that encoding input information exclusively in the intensity of the diffractive system is never sufficient for the system to be a universal function approximator. These findings enhance the understanding of the capabilities of diffractive optical systems and offer guidance for improving their training methods.

## 1   Introduction

In recent years large artificial neural networks have set new standards in research and industrial applications. The rise of these models has been largely enabled by an increase in memory and computing power. Training these large neural networks requires general purpose computing devices that are used to calculate huge matrix multiplications and other operations in parallel. Usually these devices are GPUs. However, since many artificial neural network architectures exhibit increased performance with a greater number of parameters [1], the best models are generally those with the maximum possible number of trainable parameters, only constrained by available data and training time. This leads to significant energy and computational demands during training and deployment. To tackle the ever increasing computing power requirements for large neural networks new computation architectures are needed to change the way a computer handles these large amount of floating point operations.

One promising approach for such devices is optical computing [2–4]. Optical computing approaches can be broadly divided into those using incoherent light [5, 6], and those using coherent waves [7–12]. Some of these systems rely on optical waveguides, while other systems utilize free space propagation of light [13–18]. Systems that utilize coherent free space propagation and coherent holographic plates are also referred to as diffractive deep neural networks (DDNN) [13]. An input signal is encoded in the amplitude and/or phase of a coherent wave that propagates through multiple plates, which are masks that change the phase or amplitude of a wave across its wavefront. These plates are designed in such a way, that the complete network performs a desired operation directly in the analog space of the wave. Such networks have been realized with optical light [19], terahertz waves [13] and with ultrasound [20]. They can perform a wide array of different computations, from image classification and mode conversion [18] to learning logical functions [14, 15]. These networks are inherently limited by the lack of a nonlinearity, since free space wave diffraction and hologram

38th Second Workshop on Machine Learning with New Compute Paradigms at NeurIPS 2024(MLNCP 2024).

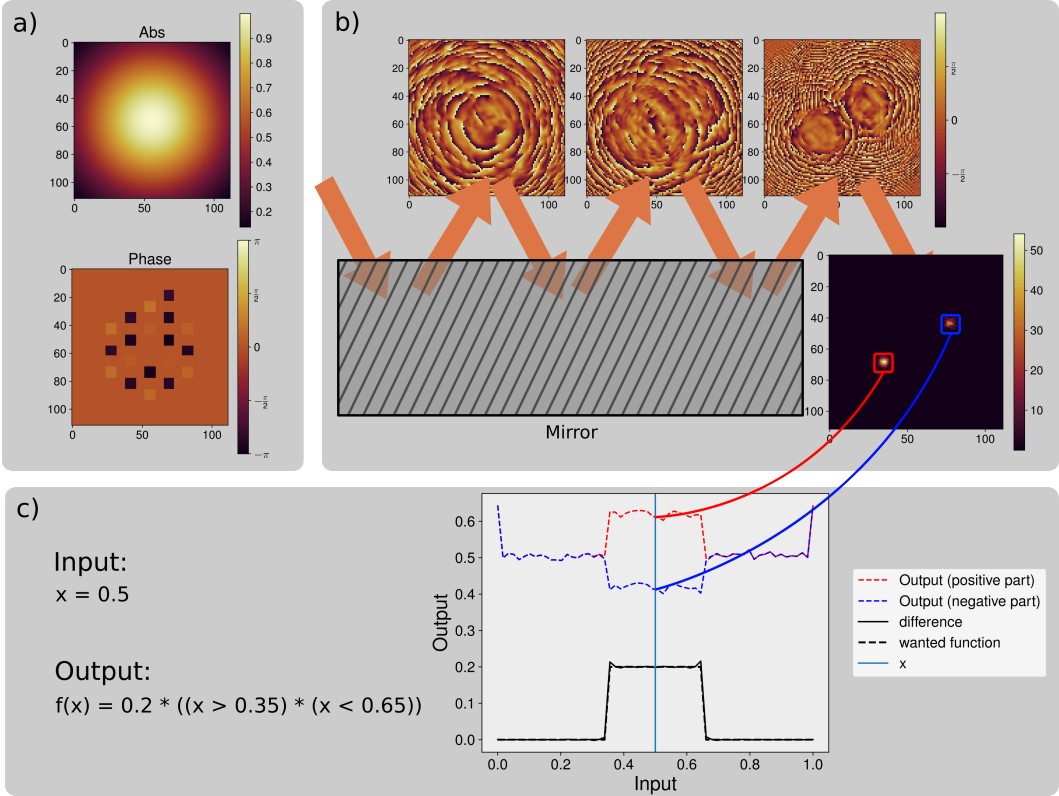

Figure 1: Overview of how a diffractive deep neural network is able to create arbitrary onedimensional functions. a) Encoding of the input ($x = 0.5$) with 20 different fixed coefficients in the phase domain of an incoming Gaussian beam. b) This encoding propagates through the network, hits 3 trained phaseplates while being reflected by a mirror and is finally captured by two regions on a CCD. c) The two regions on the CCD are subtracted and thus the result is obtained. Using this encoding the system is able to be trained to perform arbitrary functions.

plates are linear in complex space. It has been shown, that diffractive deep neural networks without a nonlinearity can perform a complex matrix multiplication [18]. Optical nonlinearities in optical neural networks are an active research field [21, 22]. However one of the easiest implementable nonlinearities is a photosensor, which maps the complex field to a voltage, that is proportional to the intensity reaching the sensor. It has numerically been shown, that such a system is able to learn certain nonlinear mappings, like an XOR logical function [15]. This raises questions about the theoretical capabilities of such systems. Here we show, that a network with a Fourier encoding is able to learn arbitrary onedimensional functions and analyze the conditions under which this is the case. This is promising for the further development of coherent optical computational systems and gives suggestions on how to best set up such a system.

## 2 Results

### 2.1 Universal function approximators

A function that fulfills the universal function approximation property can approximate any continuous function $f(x) \in C(\mathbb{R}^n)$. This means that for every $f(x) \in C(\mathbb{R}^n)$ there exists an $g(x; \theta)$ and $\theta$ such that $\max_{x \in K} |f(x) - g(x; \theta)| < \epsilon$ for any $\epsilon > 0$. $\theta$ are learnable parameters. Crucially this property does not determine how to calculate these learnable parameters, just that a combination exists, for which $f$ and $g$ are arbitrarily close everywhere in $K \subseteq \mathbb{R}^m$. Most of these theorems are used to describe artificial neural networks, and estimate bounds and limitations on the depth and width

of the layers of deep neural networks[23–25]. These theorems give us a framework to analyze the theoretical capability of diffractive deep neural network and similar systems.

## 2.2 Deep Diffractive Networks

The input of a diffractive system can be described as a vector of complex numbers, that represent the amplitude and phase of a wavefield at certain input positions.

$$I = \begin{pmatrix} I_1 e^{i\psi_1} \\ \vdots \\ I_n e^{i\psi_n} \end{pmatrix} \tag{1}$$

This input propagates through the system of diffractive plates, until it hits a photodiode at which point only the intensity, which is the square of the absolute value of the complex field, is measured.

To further derive the capability of DDNNs, we need to take a look at the output of the system. The first used to analyze the capabilities of DDNNs, is that the dimensionality of the subspace of possible transformations in a $k$ layer DDNN with an input and output pixel count of $n$ and number of weights $j$ on each plate is equivalent to $min(n^2, k*j-(k-1))$ (see [18]). Assuming a theoretically infinitely scalable input amplitude, and $k*j-(k-1) >= n^2$, this allows us to think of the propagation through a DDNN as a single matrix multiplication with complex weights.

$$Y = \mathbf{W}I \tag{2}$$

$$= \begin{pmatrix} A_{1,1} e^{i\phi_{1,1}} & \cdots & A_{1,n} e^{i\phi_{1,n}} \\ \vdots & \ddots & \vdots \\ A_{n,1} e^{i\phi_{n,1}} & \cdots & A_{n,n} e^{i\phi_{n,n}} \end{pmatrix} \begin{pmatrix} I_1 e^{i\psi_1} \\ \vdots \\ I_n e^{i\psi_n} \end{pmatrix} \tag{3}$$

.

Secondly, the measurement of the output wave only measures the intensity of the light. We can thus calculate the output for a single pixel on the photodiode as follows:

$$O_j = \left| \sum_i^n A_{i,j} I_i e^{i(\phi_{i,j} + \psi_i)} \right|^2 \tag{4}$$

$$= \sum_i^n A_{i,j}^2 I_i^2 + 2 \sum_k^n \sum_i^{k-1} A_{k,j} I_k A_{i,j} I_i \cos(\delta) \tag{5}$$

, with $\delta = \phi_{k,j} + \psi_k - \phi_{i,j} - \psi_i$. This results enables us to make further statements about the universal approximation capability of diffractive plates.

### 2.2.1 Intensity Encoding

Equation 5 enables statements about the nature of the input encoding that should be used and how it affects the output. First we analyze the case where the input is encoded only in the intensity of the incoming wave. Equation 5 shows, how the output of a single pixel depends on the input encoding **I** if the input is only encoded in the intensity of the incoming wave field

$$\mathbf{I} = \begin{pmatrix} I_1 e^{i\psi_1} \\ \vdots \\ I_n e^{i\psi_n} \end{pmatrix} = \begin{pmatrix} x_1 \\ \vdots \\ x_n \end{pmatrix} \tag{6}$$

, equation 5 reduces to

$$O_j = \sum_i^n A_{i,j}^2 x_i^2 + 2 \sum_k^n \sum_i^{k-1} A_{k,j} A_{i,j} x_k x_i \cos(\phi_{k,j} - \phi_{i,j}) \tag{7}$$

which is a second degree polynomial in $A_i x_i$, since $\cos(\phi_{k,j} - \phi_{i,j})$ is not dependent on the input encoding.

It is well known that neural networks with one hidden layer, that is of the form $y(x) = \sum_i^n c_i \sigma(\mathbf{A}\mathbf{x} - \mathbf{b})$ with $\mathbf{A}, \mathbf{x} \in \mathbf{R}^n$, $c_i \in \mathbf{R}$ and $\sigma$ a single valued function applied element-wise are universal function approximators, if and only if $\sigma$ is a not a polynomial [26].

Thus no diffractive deep neural network with intensity encoding and no other nonlinearity other than the output diode can be an universal function approximator. This result holds for the onedimensional case, as well as the multidimensional case.

### 2.3 Fourier encoding

Instead of encoding the input in the intensity, a natural next step is to encode the signal in the phase. The following encoding maps the input information multiple times with different but fixed weights. A similar method was used by Yildrim. M et. al. [27] to improve the performance of a diffractive system in image classification tasks on multiple datasets. However their methods required trainable weights that would change for new datasets. Instead of learning the input weights, we opted to simply use full integer increments to introduce higher frequencies into the system, and show that this encoding resembles the well known Fourier series. The encoding for a onedimensional variable in the optical system is displayed in equation 8

$$\mathbf{I} = \begin{pmatrix} I_1 e^{i\psi_1} \\ \vdots \\ I_n e^{i\psi_n} \end{pmatrix} = \begin{pmatrix} e^{i2\pi x} \\ e^{i4\pi x} \\ e^{i6\pi x} \\ \vdots \\ e^{in2\pi x} \end{pmatrix} \tag{8}$$

Note that all intensities are assumed to be $1$. Using this encoding in equation 5 gives

$$O_j = \sum_i^n A_{i,j}^2 + 2 \sum_k^n \sum_i^{k-1} A_{k,j} A_{i,j} cos(2\pi(k-i)x + \phi_{k,j} - \phi_{i,j}) \tag{9}$$

Higher order frequencies exist and can be used to approximate desired output functions. Due to the notable similarities to the Fourier series

$$F = C_0 + \sum_{i=1}^n C_i \cos(2\pi \frac{i}{P} x - \phi_n) \tag{10}$$

we call this encoding Fourier encoding. Since the Fourier series is able to approximate a function arbitrarily well on an interval $P$, a diffractive neural network with this encoding should be able to do the same.

However, in equation 9 the coefficients $A_i$ are all $\geq 0$, meaning that we can only create positive coefficients. The Fourier series coefficients on the other hand are $\in \mathbf{R}$. To create coefficients that can be negative, two photodiodes can be used, the first acting as a measurement device for the positive coefficients, and the second one capturing the negative coefficients. The output of these photodiodes can easily be subtracted in an analog electrical system. This addition enables the system to theoretically learn arbitrary functions.

## 3 Experimental Results

To experimentally show that a diffractive system as described above is able to learn any onedimensional function the following numerical and experimental results have been performed with a diffractive deep neural network consisting of 3 layers of $112 \times 112$ pixels on an area with a side-length of $0.1792$ mm corresponding to a pixel size of $16~\mu m$. A laser with a wavelength of $781$ nm focused in a Gaussian beam with a radius of $0.09$ mm was used as the input source. The input was encoded on a plate with the same dimensions as the trainable plates. The experimental setup used a single reflective spatial light modulator and a mirror slightly angled to reflect the beam back to the SLM, which was carefully manually calibrated. The setup is similar to the one used in [27]. This system was implemented and trained with Tensorflow [28]. Fourier encodings with 1,2,5,8,10,20,50 and 100 components have been trained to compare the influence of more complicated encodings on the capability of the system. Results for four different nonlinear functions are displayed in figure 2

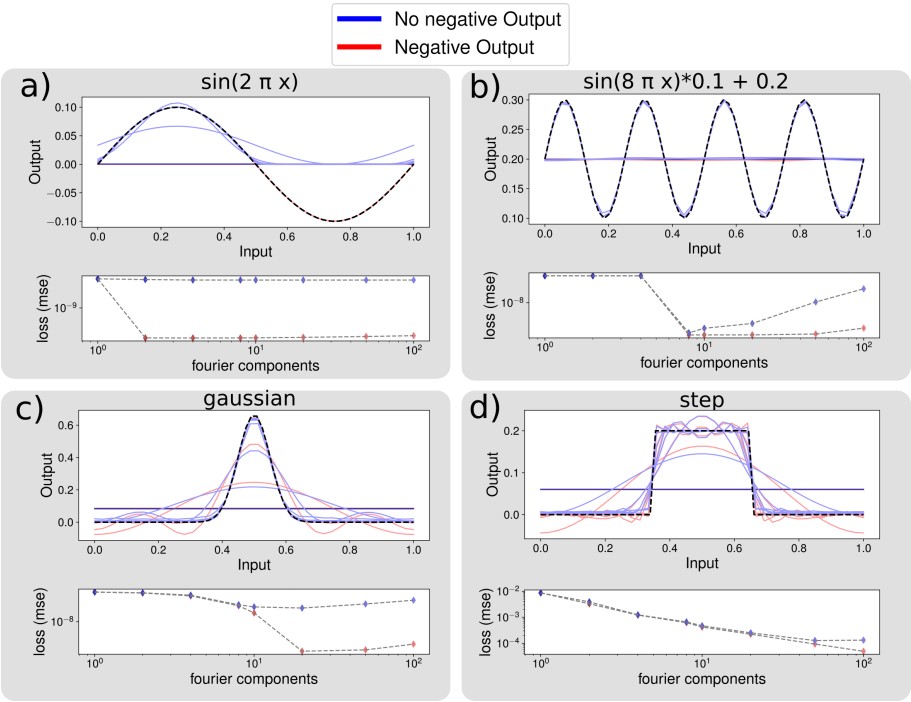

Figure 2: Numerical Results for four different nonlinear functions. The loss improves with an increasing number of Fourier components added to the optical system. a) $sin(2\pi x) * 0.1$. This function contains negative values. It cannot be learned by the system, that only considers positive output values. Only when adding negative output components can the function be fully approximated. b) $sin(8\pi x) * 0.1 + 0.2$. This function shows that multiple Fourier components are needed to approximate functions with higher frequencies c) $\frac{1}{2\sqrt{2\pi}} e^{-0.5\frac{(x-0.5)^2}{0.2^2}}$. This function needs even more Fourier coefficients to be well approximated. It can be seen, that only the system that takes negative output coefficients into account is able to approximate the function well. d) $0.2$ if $x \in [0.35, 0.65]$. This function is the hardest one to approximate, probably because it is not smooth. In all cases, the network that takes negative coefficients into account gives a better approximation.

It can be seen that the system is able to learn all four of the functions. Higher frequencies are needed to approximate all functions. This is especially clear when comparing the sinusoidal functions with increasing frequencies. While 2 frequencies are enough to train a simple sinus function, the requirements get harder with increasing frequencies. It can also be seen, that the negative output values improve the results in all cases, and are necessary to learn functions with negative values ( see plot a) of figure 2). Overall the system with negative output values manages to learn all functions to a high degree of accuracy.

These trained phase plates where tested in a physical system, and the output values where measured. For this a mirror and a single SLM was used that influenced the phase of the incoming wave. The output was measured with a CCD sensor and the two output regions where estimated and summed up. Since the output values form the camera sensor are not normed, a multiplicative offset, that corrects for the overall energy in the system and an additive offset that corrects for the base noise level in the camera sensor where manually chosen for all measured output values. The results are displayed in figure 3 The results confirm the numerical experiments. They show that it is possible to train a diffractive deep neural network to perform arbitrary nonlinear functions.

## 4   Discussion

We have shown, that a coherent optical system with diffractive plates, also called deep diffractive neural network, is capable of performing arbitrary nonlinear functions in one dimension. We utilized an encoding, that embeds the input variable into multiple phase inputs, that have different factors,

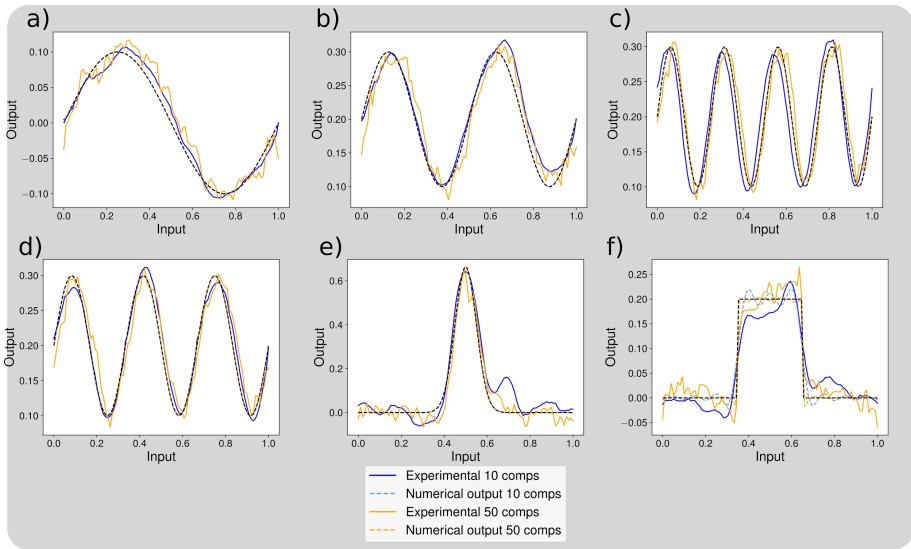

Figure 3: Real world experimental results. The functions for the plots a) to f) are the same was in figure 2. It can be seen, that the measured functions match the numerical ones. Some noise is introduced during measurement, due to the nature of an analog system.

that match those of the Fourier series. Furthermore we have shown a mathematical proof that a diffractive system that solely relies on intensity inputs for the data can never be an universal function approximator.

The experiments support the theoretical findings. Multiple nonlinear functions using the Fourier encoding have been measured on a real world system after training them on a backpropagatable simulation. A negative output region was useful for all trained functions, but made the biggest difference for functions that had negative values for obvious reasons.

One additional question is if a system that encodes the information in the phase without using multiple input factors, like the Fourier encoding can potentially be considered a universal function approximator. Using an input encoding of

$$\mathbf{I} = \begin{pmatrix} I_1 e^{i\psi_1} \\ \vdots \\ I_n e^{i\psi_n} \end{pmatrix} = \begin{pmatrix} e^{ix_1} \\ \vdots \\ e^{x_n} \end{pmatrix} \tag{11}$$

leads to an output of

$$O = O^+ - O^- = \sum_j^m c_i(1 + cos(\Delta_{\phi_1,\phi_2} - x)) \tag{12}$$

which is equivalent to a neural network with fixed weights of 1 and a bias $\Delta_{\phi_1,\phi_2}$ in the first layer, a cosine activation function, and trainable weights in the second layer. Theoretical results exist, that show that these networks can potentially be universal function approximators [29]. However we have been unable to train arbitrary nonlinear functions using only this encoding, without inputs that introduce higher frequencies.

The insights gained about diffractive systems should also be helpful when training DDNNs for different tasks, that are potentially more demanding than a onedimensional function. However since the input data is multiplied for every new Fourier coefficient added to the system, spacial demands on the input encoding are potentially a concern. In general the results highlight a few key things that should be taken into account when training DDNNs. First, the information should not be encoded solely in the intensity of the incoming wave, to ensure cosine nonlinearities in the output, potentially using more than one frequency per input value. Secondly, a region with negative values has been shown to be important for all functions. Such a negative output region can be realized with an analog electrical system, consisting of two photodiodes. Obeying by these results should lead to improved performance of any free space diffractive system for information processing purposes.

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
