# OpenReview forum: "Universal approximation capabilities of coherent diffractive systems"
_NeurIPS.cc/2024/Workshop/MLNCP — MLNCP Poster_

### Official Review · Reviewer_WKEt · 2024-10-03
**Universal approximation capabilities of coherent diffractive systems**

**Rating:** 3
**Confidence:** 4

**Review:**

The manuscript “Universal approximation capabilities of coherent diffractive systems” analyzes the ability of a deep diffractive neural network (DDNN) to act as a universal function approximator. The authors claim that a DDNN system containing multiple phase plates and two output photodetectors is theoretically capable of learning any one-dimensional function. However, I do not agree with the authors’ claim.

It is well known that a single hidden layer feedforward network with a non-polynomial activation function can be a universal function approximator. However, this requires that the elements in both “weight” matrices can be set as any real value. The authors state (line 93-94) that Eq. 9 "is equivalent to a single layer neural network with fixed weights of 1 in the first layer and offsets that can be trained." Thus, since the (equivalent) first layer is not trainable, I do not see how the DDNN can be a universal approximator.  The authors are also unable to "demonstrate experimentally that they can approximate functions that contained higher order frequencies, or where different from the family of cosine functions."

Additionally, the weight matrix representing the wave’s propagation in Eq. 3 with elements $Ae^{i\phi}$ cannot be set as any arbitrary value. DDNNs consist of a series of free-space propagation and element-wise multiplication operations. The authors provide no proof that they can tune a DDNN to represent an arbitrary matrix. As an example, it is clearly not possible to represent a transformation with a matrix norm that is larger than 1, because this would imply that energy is added to the system during the transformation.

Due to the flawed reasoning in the theory and lack of rigorous analysis, I cannot recommend this submission for acceptance to the workshop.

Minor remarks:

- There are several grammar and punctuation mistakes throughout the paper (“wavefrong “ in line 33, unnecessary comma in lines 70, 73, etc). I encourage the authors to further proof-read their manuscript before submission.

---

### Decision · Program_Chairs · 2024-10-10

Accept (Poster)